# Protein status of people with phenylketonuria: a scoping review protocol

Sarah Firman,[1,2] Radha Ramachandran,[2] Kevin Whelan,[1] Oliver C Witard,[1,3] Majella O'Keeffe[1,4]

¹Department of Nutritional Sciences, King's College London, London, UK
²Adult Inherited Metabolic Diseases, Guy's and St Thomas' NHS Foundation Trust, London, UK
³Centre for Human and Applied Physiological Sciences, King's College London, London, UK
⁴School of Food and Nutritional Sciences, University College Cork, Cork, Ireland

**Correspondence to**
Dr Majella O'Keeffe;
majella.okeeffe@ucc.ie

## ABSTRACT

**Introduction** Phenylketonuria (PKU) is a disorder of protein metabolism resulting in an accumulation of phenylalanine in the body. Dietary management consists of altering the sources of ingested protein to limit phenylalanine intake. Current dietary protein guidelines for PKU are based on limited scientific evidence, thus it remains unclear whether current practice leads to optimal protein status in people with PKU. To date, no attempt has been made to systematically evaluate the protein status of people with PKU, using a combination of validated anthropometric, biochemical and functional measurement tools. Furthermore, factors known to influence protein status in the general population warrant consideration when determining protein status in individuals with PKU, alongside factors unique to PKU such as the type of protein substitute consumed. Understanding the impact of these variables on protein status is crucial to developing a personalised approach to protein recommendations for optimising health and functional outcomes in people with PKU. Therefore, the aim of this scoping review is to examine existing evidence regarding the protein status of people with PKU, and to investigate the nutritional and lifestyle variables that influence protein status.

**Methods and analysis** This review will be guided by Arksey and O'Malley's framework, along with guidance from Levac *et al*, Pawliuk *et al* and the Joanna Briggs Institute. The following databases will be searched: MEDLINE (Ovid), Embase, CENTRAL, Web of Science and Scopus, alongside grey literature. Identified literature will be assessed by two independent reviewers for inclusion. Descriptive numerical analysis will be performed and a narrative summary will accompany the tabulated results describing how study findings relate to the review questions.

**Ethics and dissemination** This review protocol does not require ethical approval. Findings will be disseminated through peer-reviewed publication, presented at relevant conferences, and shared with a patient research advisory group to inform discussions on future research.

## Strengths and limitations of this study

► This review is the first to collate data from various anthropometric, biochemical and functional measurements of protein status in people with phenylketonuria (PKU).

► With limited research in the area of PKU, the broad research question will allow for the mapping of available evidence, and to identify the gaps in existing research to inform the direction of further research into improving the care provided for people with PKU.

► The search strategy includes identification of published and unpublished literature, including sourcing grey literature by contacting specialist groups and hand searching conference and meeting reports.

► The protocol includes a clearly defined inclusion criteria aligned with Joanna Briggs Institute Population, Concept and Context strategy to ensure transparency of information sourced for evaluation in the review.

► Aligned with the accepted framework for scoping reviews, this study does not include formal assessment of the quality of evidence.

and irreversible intellectual disability, in addition to reducing other neurological consequences associated with high phenylalanine throughout life. Lifelong treatment is recommended for individuals with blood phenylalanine levels above specific phenylalanine targets, which can vary depending on the management guidelines adopted.[1–3] Treatment should be started according to country-specific guidelines or, where guidelines are not available, to European or US consensus guidelines.[1–3] Currently, dietary management provides the mainstay of treatment and consists of altering the sources of ingested protein to limit phenylalanine intake, and supplementation with phenylalanine-free or low-phenylalanine protein sources to meet protein requirements.[4] In practice, this dietary regimen involves restricting protein intake from food and instead providing the majority

## INTRODUCTION

Phenylketonuria (PKU; OMIM 261600) is an inborn error of protein metabolism that leads to an accumulation of phenylalanine in the blood and brain. Early diagnosis and treatment are essential in preventing profound

BMJ

(up to 80%) of protein intake from phenylalanine-free or low-phenylalanine protein substitutes, consisting of either L-amino acids (L-AA) or casein glycomacropeptide (CGMP),[5][6] and more recently, protein substitutes using prolonged-release amino acid technology.[7–9]

A key limitation of existing guidelines on dietary protein requirements for people with PKU[1–3] is that the data has been extrapolated from studies that estimated protein requirements in healthy populations.[4] These studies used nitrogen balance methodology to evaluate the minimum protein (ie, nitrogen) intake required to balance nitrogen losses at the whole-body level of physiology.[10] In contrast to the definition of a protein requirement, a protein recommendation serves to optimise metabolic function and improve health and functional outcomes in a given population group.[11][12] Experimental studies conducted in healthy young (18–35 year old) adults without PKU provide useful insight into the impact of L-AA supplementation on plasma amino acid kinetics.[13–17] The rapid rise in plasma amino acid concentrations observed with ingestion of L-AA has been shown to result in greater amino acid oxidation rates and lower protein retention at the whole-body level in comparison with whole protein. To account for the reduced uptake and utilisation of amino acids from protein substitutes, current advice is to apply a correction factor of 20%–40% in excess of the protein requirement guidelines of the general population.[1–3] However, the evidence underpinning protein requirement guidelines for dietary management of adults with PKU is limited, and in their current form may not adequately offset age-related changes in protein metabolism across the adult lifespan in people with PKU.[4]

In addition to total daily protein intake, multiple factors are known to influence protein status in the non-PKU population. These factors include, but are not limited to, the protein dose ingested per meal/serving, protein source, the timing of protein intake in relation to physical activity and/or other nutrients, the daytime distribution pattern of protein intake and also the co-ingestion of other nutrients alongside protein.[18–24] Furthermore, age, sex and physical activity status are all known to modify the metabolic fate of ingested protein.[23–27] These variables are also relevant when considering research undertaken

to determine protein status in individuals with PKU. In addition, there are factors unique to the dietary regimen of people with PKU that warrant consideration such as type of protein substitute (L-AA vs CGMP vs protein substitutes using prolonged release technologies), dose and timing of protein substitute ingestion, the proportion of total protein intake from dietary protein versus protein substitutes, adherence to protein substitute prescription and the period of time spent on and off a phenylalanine restricted diet.

It remains unclear whether existing protein requirement guidelines for people with PKU translate to optimal protein status across their lifespan. Developing an understanding of the variables that modulate protein status, both positively and negatively, will be crucial to informing future research on personalising protein recommendations for individuals of all ages with PKU, with the intention of improving the care provided for people with PKU. The term 'protein status' encompasses a vast range of measurements and physiological outcomes, and for the purpose of this scoping review the term will include those parameters detailed in table 1. Studies investigating body composition and biochemical measurements of protein nutritional status have been conducted in children and adults with PKU,[28–34] yet no contemporary synthesis has been undertaken. Mapping the available evidence is important to identify the direction(s) for future research into establishing evidence-based protein recommendations for dietary management of PKU. In this regard, a scoping review provides the ideal approach to explore the breadth of literature on the protein status of people with PKU in order to map and synthesise the evidence and identify the key variables that influence protein status in this population.

To date, no attempt has been made to systematically evaluate the protein status of people with PKU, using a combination of validated anthropometric, biochemical and functional measurement tools. Although three systematic reviews of protein status in people with PKU are currently registered on PROSPERO, two are exclusively focused on bone health (which is not the focus of the proposed review), with the other systematic review and meta-analysis investigating measurements of body composition only, in patients with PKU.[35] Our scoping review

| Table 1 | Measurements of protein status considered in the review |
|---|---|
| Anthropometric | Body composition (fat free mass, lean body mass and / or skeletal muscle mass) via dual-energy X-ray absorptiometry, bioelectrical impedance analysis, total body electrical conductivity, BodPod whole body air-displacement plethysmography or skinfolds. |
| Biochemical | 3-methylhistidine concentrations, albumin, prealbumin, transthyretin, retinol-binding protein, urea production, blood urea nitrogen, urinary nitrogen, total body nitrogen, whole-body protein metabolism, and plasma amino acids concentrations, urea production and creatinine (where the author(s) have specifically used these as a measure of protein status). |
| Functional | Hand-grip strength, the Short Physical Performance Test (including tests of balance, gait speed and timed sit-to-stand), one-repetition max (or a five-repetition max for older adults) and $VO_2$max testing (or $VO_2$peak for older adults) and other validated measures of muscle function (ie, isokinetic quadriceps strength using dynamometry and vertical jump performance using force platform technology). |

is unique in taking a holistic approach by encompassing a range of validated measurements of protein status, including biochemical (eg, 3-methylhistidine concentrations, prealbumin and urinary nitrogen) and functional parameters (eg, hand-grip strength and the Short Physical Performance Test), in addition to body composition. Furthermore, our review is the first to investigate the key variables that influence protein status in a PKU cohort across the lifespan. A scoping review is pertinent to our research questions as it enables the breadth of evidence to be mapped and synthesised qualitatively, and to identify gaps in current evidence where future research can focus with the intention to ultimately improve patient care.

## Objective

The primary objective of this scoping review is to examine existing evidence regarding the protein status of people with PKU and identify key nutritional and lifestyle variables that influence protein status in people with PKU.

## METHODS AND ANALYSIS
### Protocol design

The protocol of the proposed scoping review was informed by Arksey and O'Malley's framework involving the following five stages: (1) identifying the research question, (2) identifying relevant studies, (3) study selection, (4) charting the data, (5) collating, summarising and reporting the results,[36] with consideration for Levac *et al's* methodological enhancements,[37] guidance by Pawliuk *et al*[38] and the Joanna Briggs Institute (JBI) methodology for scoping reviews.[39]

### Stage 1: identifying the research question

Through consultation with the research team, the key elements of the JBI PCC mnemonic (Population, Concept and Context) were established for this review (see eligibility criteria).[39] These elements are reflected in the main research question and informed the inclusion and exclusion criteria.

The main research question for this scoping review is: What is the existing evidence of the protein status of people with PKU across the lifespan? For this review, the assessment of protein status encompasses biochemical, functional and anthropometric measurements, as outlined in table 1.

In addition to this main question, the review will focus on the following subquestions:
1. How is protein status assessed in people with PKU?
2. Which variables, known to influence protein status in the general population, have been investigated when assessing protein status in people with PKU?
3. Which variables, known to influence protein status in the general population, modulate protein status in people with PKU?

### Stage 2: identifying relevant studies
#### Information sources

The following databases will be searched to identify relevant literature: MEDLINE (Ovid), Embase (Ovid), CENTRAL, Web of Science and Scopus. Conference proceedings and abstracts sited in Embase (Ovid) will be examined. Unpublished studies and grey literature will be sought by contacting experts in inherited metabolic diseases specialist groups and sourced from reports available from the Society for the Study of Inborn Errors of Metabolism (Journal of Inherited Metabolic Disease) and International Congress of Inborn Errors of Metabolism (Journal of Inborn Errors of Metabolism and Screening) international meetings from 2010 to 2020. Hand searching of the reference lists of articles selected for full-text review will be undertaken to identify any additional literature. Articles published in English only will be included. No date restriction will be applied to the searches. The search to identify relevant literature will be conducted between June and July 2021.

#### Search strategy

A preliminary search was undertaken by the research team to identify key words in titles and abstracts from relevant literature, and along with the index terms, were used to develop a full list of 96 search terms to encompass 'protein status'. From this, a list of 19 key terms were agreed on by the research team to be included in the full search strategy. The search strategy was developed with guidance from a librarian and piloted in MEDLINE (Ovid) by the research team, see online supplemental material.

#### Eligibility criteria

As outlined in the JBI methodology for scoping reviews,[39] establishing clear inclusion criteria is important in defining the scope of the review and to guide the research team when identifying relevant studies to include. The JBI PCC strategy was used to develop the following inclusion and exclusion criteria:

#### Population

This scoping review will consider studies that include participants diagnosed with PKU. Studies will be excluded if the study population incorporates the following: (1) women with PKU who are pregnant, as dietary management and phenylalanine targets can differ during preconception, pregnancy and postpartum; and (2) people with other comorbidities that could influence protein intake, protein digestion and amino acid absorption, such as pancreatic insufficiency, coeliac disease, irritable bowel syndrome, inflammatory bowel disease, diabetes, cancer and/or having had a gastric bypass. All age categories will be considered and defined as 'children' <18 years of age, 'adults' ≥18 years of age and 'older adults' >60 years of age.

#### Concept

This scoping review will evaluate studies that report the biochemical, functional and/or anthropometric measurements of protein status in people with PKU, as outlined in table 1. Where analysis of overall nutritional status has been reported, only measurements of protein status will

be considered. Studies that report dietary protein intake without protein status outcome measurements, or those reporting only body weight, height and body mass index, will be excluded.

As detailed previously, numerous variables influence protein status in the general population, including total daily protein intake, protein dose consumed on a per meal basis, the daytime distribution pattern of protein intake and in relation to exercise, and protein source, in addition to age, sex and physical activity status; all of which warrant consideration when assessing protein status in PKU. Furthermore, factors relevant to PKU dietary management such as the proportion of total protein intake from dietary protein versus protein substitutes, type of protein substitute (L-AA vs CGMP vs prolonged release technologies), adherence to protein substitute prescription and metabolic control warrant consideration. An evaluation of whether these variables have been controlled for by the relevant studies, and the impact of these variables on protein status in people with PKU, will be included in this review.

Although lifelong treatment is recommended in the management of PKU, some adolescents and adults may choose to have periods of time on an unrestricted diet. If available, information on participants' history with the PKU diet will be obtained, as the impact of changes in total protein intake on and off diet on protein status warrants consideration. Socioeconomic status indicators will also be extracted.

### Context
This review will consider studies that report protein status in people with PKU from an international perspective. Although the existing literature base on PKU is relatively limited given that PKU is a rare condition, the inclusion of all geographical areas will ensure all available evidence

can be considered. Dietary management differs between countries, and therefore consideration will be given to the protein requirements and management guidelines specific to each population included in the review.

### Stage 3: study selection
The study selection will be undertaken in accordance to the Preferred Reporting Items for Systematic Reviews and Meta-Analyses extension for scoping reviews (PRISMA-ScR) flow diagram[40] (see figure 1) and presented in the final scoping review. Following the search, all identified records will be uploaded into Zotero V.5.0 (George Mason University, USA) and duplicates removed automatically and manually. The review process will consist of two stages. First, titles and abstracts will be screened by two independent reviewers to determine potential eligibility. Second, the full text of records deemed relevant will be retrieved and assessed in detail against the inclusion criteria by the same two independent reviewers. As recommended by Pawliuk et al,[38] the study selection process will be tested using 5% of the articles identified, prior to initiating the formal scoping review study selection, to ensure the reviewers understand the inclusion criteria. The two reviewers will meet to resolve any disagreements that arise at each stage of the study selection process, and where consensus cannot be reached, through discussions with a secondary review panel.

### Stage 4: charting the data
Data will be charted by two independent reviewers using an adapted version of the JBI extraction instrument.[39] The key data to be extracted from the selected articles are outlined in box 1 and will include details about the PCC and information relevant to the review questions. The draft data charting tool will be piloted for purpose, and modified as required. Once the first five articles have

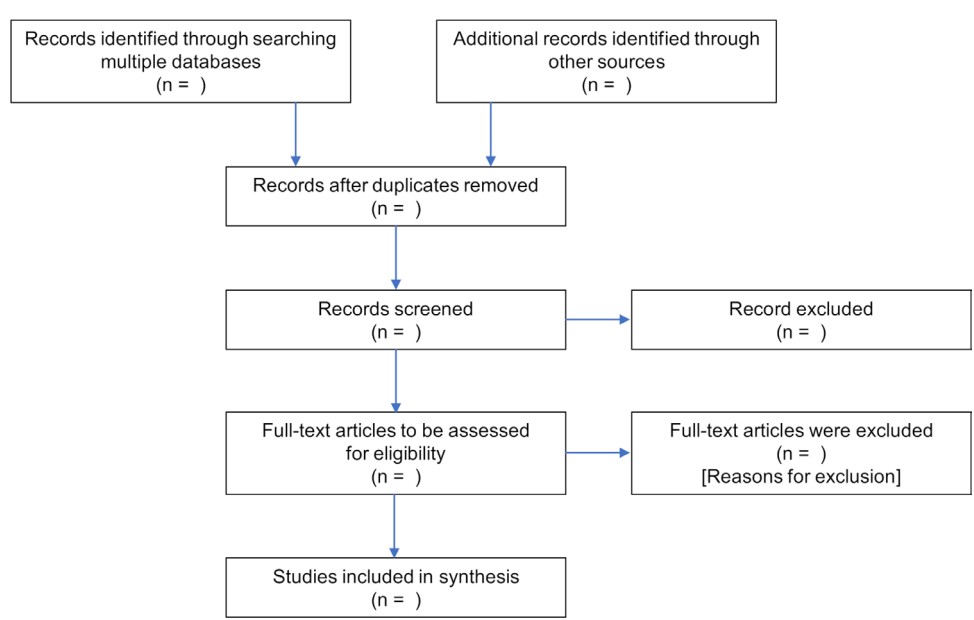

**Figure 1** Preferred Reporting Items for Systematic Reviews and Meta-Analysis flow diagram for scoping review process. Retrieved from Tricco *et al*.

## Box 1 Key data to be extracted

1. Author(s).
2. Year of publication.
3. Title and journal.
4. Country of origin.
5. Aims/purpose.
6. Population characteristics (sample size, age, sex, ethnicity, time of diagnosis, metabolic control, genotype, socioeconomic status).
7. Methodology/methods.
8. Intervention type, comparator and details of these (if applicable).
9. Duration of the intervention (if applicable).
10. Primary outcome.
11. Protein status outcomes and how these were measured.
    a. Anthropometric.
    b. Biochemical.
    c. Functional.
12. Variables controlled for in the study.
13. Key findings that relate to the review questions.
14. Study limitations.

been charted by the two reviewers independently, the reviewers will meet to ensure there is consistency in information extracted.

### Stage 5: collating, summarising and reporting the results

The extracted data will be tabulated in a manner that aligns with the research objectives and questions of this scoping review. This table will provide a descriptive numerical summary of the study characteristics including the total number of studies involved, year of publication, study design, study population characteristics and primary outcomes. A further descriptive numerical summary will outline the measurements of protein status used and the findings from the studies (including any descriptive statistics and effect size where available). Due to existing concerns regarding whether current guidelines on protein requirements are adequate to offset age-related changes in protein metabolism across the adult lifespan,[4] in addition to reporting the results on protein status across the lifespan, a summary of evidence from research specific to adults and older adults will be included.

To address the subquestions on the variables considered and the impact of variables on protein status, the variables controlled for in each study will be tabulated alongside the study's results of protein status to map findings of any correlations of these variables with protein status in people with PKU. This exercise will identify gaps in current research where variables known to influence protein status in the general population need further consideration when determining protein recommendations for PKU dietary management guidelines. A narrative summary will accompany the tabulated results and will describe how the results relate to the review's objective and questions. The PRISMA-ScR will guide the reporting of the results.[40]

### Patient and public involvement

There is no patient and public involvement in the design or conduct of the review. The review findings will be shared with a patient and public involvement group, established to advise on research in PKU, to inform discussions for the direction of future research.

## ETHICS AND DISSEMINATION

As this study is based on the review of publicly available information, the review does not require ethical approval. This review is the first step in a larger research project to contribute to the development of robust evidence-based protein recommendation guidelines for individuals with PKU. Findings from this review will be disseminated through peer-reviewed publication, and shared at conferences and national meetings on inherited metabolic disease. As outlined in 'Patient and public involvement', the review will also be shared with a patient group established to advise on research in PKU to inform plans for further research.

**Acknowledgements** The Library Services at King's College London for their guidance in the development of the search strategy.

**Contributors** SF, RR, KW, OW and MOK were involved in conceptualisation, development of the research questions and design of protocol methodology including the search strategy. SF prepared the original draft. MOK was lead supervisor for the development of the protocol. RR, KW, OW and MOK provided critical revision of the draft. SF, RR, KW, OW and MOK revised and approved the final manuscript.

**Funding** SF is funded by Health Education England (HEE)/National Institute for Health Research (NIHR300395). This review presents independent research funded by the HEE/NIHR. The views expressed are those of the author(s) and not necessarily those of HEE, the National Health Service, the NIHR or the Department of Health and Social Care.

**Competing interests** SF has received funding to attend conferences and study days from Nutricia, Vitaflo International and Dr Schär UK Ltd, and consulting fees from Vitaflo International and Meta Healthcare. KW is in receipt of research funding from Danone and has acted as a consultant for Danone.

**Patient consent for publication** Not required.

**Provenance and peer review** Not commissioned; externally peer reviewed.

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
