## [Reviewer comments · BMJ Open]

ARTICLE DETAILS

TITLE (PROVISIONAL)	Protein status of people with phenylketonuria: A scoping review protocol
AUTHORS	Firman, Sarah; Ramachandran, Radha; Whelan, Kevin; Witard, Oliver; O'Keeffe, Majella

VERSION 1 – REVIEW

REVIEWER	Li, Hong Sichuan University West China Second University Hospital, National Office for Maternal and Child Health Surveillance of China
REVIEW RETURNED	30-Mar-2021

GENERAL COMMENTS	This scoping review aims to examine existing evidence about the protein status of people with PKU and the relative influence factors. The results of this study will show us the current research progress in this field, which helps us to identify gaps in current evidence. This proposal is well written with clear definition of research questions, reasonable and actionable research procedure. Minor comments: 1. Because the quality of evidence will not be assessed, it might be difficult to determine whether the results of the study are robust or extrapolatable.2. If the relative research evidence about children with PKU is enough, it is suggested that children should be divided into different age groups, such as < 1 yr, 1-3 yrs, 3-6 yrs, 6-17+ yrs. The influencing factors of protein status of children with PKU might be very different in different age groups.
---

REVIEWER	Yang, Feng-Jung National Taiwan University Hospital, Internal Medicine - Rare Diseases Center, Medical Genetics
REVIEW RETURNED	02-Apr-2021

GENERAL COMMENTS	The protein and other PKU specific supplements should be considered at the same time.
---

REVIEWER	Rocha, Júlio César Universidade Nova de Lisboa, Nutrition and Metabolism
REVIEW RETURNED	12-Apr-2021

GENERAL COMMENTS	Congratulations to the authors for this interesting review protocol. It is focused on an interesting topic that really deserves further and deeper attention.
---

	Anyway, there are some questions that can be raised: 1. The first paragraph of the introduction refers "... country-specific phenylalanine targets...". Can this be applicable to all countries since there are countries where recommendations/guidelines are not available? I would suggest adding also a sentence suggesting that treatment should be started according to country-specific or to European or to US guidelines. In addition, it is strange that the first paragraph does not have any reference.2. In the 3rd paragraph of introduction, when the " type of protein substitute " is discussed, I think it is also necessary to discuss the new technological approaches already available in the market in some countries, using slow release technologies. There are already some references in the literature discussing the impact of these new protein substitutes.3. Still, on the introduction the authors refer: "Studies investigating body composition and biochemical measurements of protein nutritional status have been conducted in children and adults with PKU[26–30],". This list of publications [26-30] is really incomplete and does not reflect the current status of knowledge in PKU.4. Do authors think that hand-grip strength will be a useful methodology to discriminate muscular strength in PKU patients, considering the absence of evidence of undernutrition in these individuals?5. Limiting the time frame for unpublished studies "international meetings from 2015-2020" may have additional risks. In the last years, with new pharmacological treatments the research focusing protein status may have been reduced, so why not to extend the period at least 5 additional years back?6. Population criteria: what does it mean "people with raised phenylalanine levels not related to phenylalanine hydroxylase deficiency"? Do authors think that it would be beneficial to have also criteria about metabolic control of patients included in the studies, together with information about being early / late diagnosed?7. The authors refer "For the purpose of this review, 'on diet' is defined as restricting dietary protein and/or taking at least one protein substitute daily, and 'off diet' is defined as eating freely without restriction and not taking protein substitutes". How do authors think they will be able to precisely identify the patients correctly, since there are frequent changes on dietary compliance in most, if not all, patient's series? In addition, do authors think they will have access to this data?
--	--

VERSION 1 – AUTHOR RESPONSE

Reviewer: 1

Prof. Hong Li, Sichuan University West China Second University Hospital

Comments to the Author:

This scoping review aims to examine existing evidence about the protein status of people with PKU and the relative influence factors. The results of this study will show us the current research progress in this field, which helps us to identify gaps in current evidence. This proposal is well written with clear definition of research questions, reasonable and actionable research procedure.

RESPONSE: We would like to thank reviewer 1 for taking the time to review our manuscript and for their encouraging comments.

Minor comments:

1. Because the quality of evidence will not be assessed, it might be difficult to determine whether the results of the study are robust or extrapolatable.

RESPONSE: Thank you for your comment. As highlighted by the reviewer above, the focus of this review is to map and summarise the evidence to identify gaps for further research, and therefore a scoping review has been selected as the most appropriate method. We acknowledge this study will not include formal assessment of the quality of evidence. However, we have taken an approach that is based on the accepted framework for scoping reviews.

2.If the relative research evidence about children with PKU is enough, it is suggested that children should be divided into different age groups, such as < 1 yr, 1-3 yrs ,3-6 yrs, 6-17+ yrs. The influencing factors of protein status of children with PKU might be very different in different age groups.

RESPONSE: Thank you for your suggestion. We agree that the factors influencing protein status could be different for children from birth to 17+ years of age. Whilst the evidence based in children with PKU is larger than adults, due to the rare nature of PKU, the studies tend to group age cohorts to increase the sample sizes and therefore it may be challenging to determine the specific influencing factors from the studies available for different age groups from 0-17 years. However, as you have suggested, age within the 0-18 years old cohort is an important consideration. Therefore, we have updated paragraph 2 under 'Concept' (page 8) to reiterate that 'age' is a factor of interest when assessing protein status in PKU: '... numerous variables influence protein status in the general population, including total daily protein intake, protein dose consumed on a per meal basis, the daytime distribution pattern of protein intake and in relation to exercise, and protein source, in addition to age, sex and physical activity status; all of which warrant consideration when assessing protein status in PKU.'

Reviewer: 2

Dr. Feng-Jung Yang, National Taiwan University Hospital

Comments to the Author:

The protein and other PKU specific supplements should be considered at the same time.

RESPONSE: We would like to thank reviewer 2 for taking the time to review our manuscript and for your comment. We agree that the total protein intake as well as the proportion of protein from natural/dietary protein and from PKU supplements are key factors that warrant consideration throughout this scoping review on the protein status of people with PKU. We also agree that type of PKU supplement needs to be considered and, in addition to L-amino acids (L-AA) and casein glycomacropeptide (CGMP), we have now updated the manuscript to include protein substitutes utilising the newer prolonged release amino acid technology.

We have updated paragraph 2 under 'Concept' (page 8) to specify: 'Furthermore, factors relevant to PKU dietary management such as the proportion of total protein intake from dietary protein versus protein substitutes, type of protein substitute (L-AA versus CGMP versus prolonged release technologies), adherence to protein substitute prescription, and metabolic control warrant consideration.'

Reviewer: 3

Prof. Júlio César Rocha, Universidade Nova de Lisboa, Centro Hospitalar de Lisboa Central EPE

Comments to the Author:

Congratulations to the authors for this interesting review protocol.

It is focused on an interesting topic that really deserves further and deeper attention.

RESPONSE: We would like to thank reviewer 3 for taking the time to review our manuscript and for their positive comments and encouragement. We believe this is an important topic in optimising health outcomes of those with PKU, and it is reassuring to know that the reviewer also considers it so.

Anyway, there are some questions that can be raised:

1. The first paragraph of the introduction refers "... country-specific phenylalanine targets...". Can this be applicable to all countries since there are countries where recommendations/guidelines are not available? I would suggest adding also a sentence suggesting that treatment should be started according to country-specific or to European or to US guidelines. In addition, it is strange that the first paragraph does not have any reference.

RESPONSE: Thank you for your suggestions on the introduction. As requested, we have reworded this section to outline that phenylalanine targets can vary depending on the management guidelines adopted, and have included your suggestion on a sentence regarding treatment:

'Lifelong treatment is recommended for individuals with blood phenylalanine levels above specific phenylalanine targets, which can vary depending on the management guidelines adopted[1-3]. Treatment should be started according to country-specific guidelines or, where guidelines are not available, to European or United States consensus guidelines[1-3].'

The first paragraph has been updated with references.

2. In the 3rd paragraph of introduction, when the " type of protein substitute " is discussed, I think it is also necessary to discuss the new technological approaches already available in the market in some countries, using slow release technologies. There are already some references in the literature discussing the impact of these new protein substitutes.

RESPONSE: Thank you for your suggestion. We agree that it is important to also include protein substitutes that use the prolonged release amino acid technologies. We have updated both the 1st and 3rd paragraph of the introduction accordingly:

(Paragraph 1) In practice, this dietary regimen involves restricting protein intake from food and instead providing the majority (up to 80%) of protein intake from phenylalanine-free or low-phenylalanine protein substitutes, consisting of either L-amino acids (L-AA) or casein glycomacropeptide (CGMP)[5,6], and more recently, protein substitutes utilising prolonged-release amino acid technology[7–9].

(Paragraph 3) In addition, there are factors unique to the dietary regimen of people with PKU that warrant consideration such as type of protein substitute (L-AA versus CGMP versus protein substitutes utilising prolonged release amino acid technologies)....

3. Still, on the introduction the authors refer: "Studies investigating body composition and biochemical measurements of protein nutritional status have been conducted in children and adults with PKU[26–30]". This list of publications [26-30] is really incomplete and does not reflect the current status of knowledge in PKU.

RESPONSE: Thank you for your comment on this. We have updated the references cited to include a broader range of publications to reflect the current evidence available in both adults and children with PKU.

4. Do authors think that hand-grip strength will be a useful methodology to discriminate muscular strength in PKU patients, considering the absence of evidence of undernutrition in these individuals?

RESPONSE: Thank you for your question on this methodology. There has been limited research investigating functional outcomes in PKU. Hand-grip strength is a measure of maximum force created by the forearm muscles and is commonly used as an assessment tool for the measurement of overall strength. Grip strength was used in the study by Choukair et al (2017)

(<https://pubmed.ncbi.nlm.nih.gov/27878409/>) to investigate functional muscle-bone in adolescents and adults with PKU. This study found that muscle grip force was significantly reduced in their PKU cohort and also highlighted that the regression line slope for the correlation between muscle cross-sectional area and muscle force was less steep for patients with PKU compared to the reference population. Therefore, whilst muscle mass might be similar between individuals with PKU and healthy age-matched controls, functional outcomes may still differ.

As highlighted above, hand-grip strength is the most accepted and published measurement of whole body muscular strength. The assessment of grip strength requires minimal equipment, space and labour and thus, in practice, can be easily adopted in the clinical setting. Hence, we feel that grip strength should serve as the primary outcome measurement for the assessment of muscle functional status. Alongside other assessments of functional status, these measurements will help inform useful methods for measuring functional outcomes in individuals with PKU.

5. Limiting the time frame for unpublished studies “international meetings from 2015-2020” may have additional risks. In the last years, with new pharmacological treatments the research focusing protein status may have been reduced, so why not to extend the period at least 5 additional years back?

RESPONSE: Thank you for this suggestion. You raise an important point about the research focus in the last few years. As requested, we have updated the manuscript to include unpublished studies from 2010-2020.

6. Population criteria: what does it mean “people with raised phenylalanine levels not related to phenylalanine hydroxylase deficiency”? Do authors think that it would be beneficial to have also criteria about metabolic control of patients included in the studies, together with information about being early / late diagnosed?

RESPONSE: We had specified “people with raised phenylalanine levels not related to phenylalanine hydroxylase deficiency” as part of the exclusion criteria, meaning those with tetrahydrobiopterin (BH4) deficiency (eg DHPR deficiency) would be excluded. However, as the protocol specifies ‘This scoping review will consider studies that include participants diagnosed with PKU’, we have realised that it is not necessary to specify this exclusion criteria and have therefore removed it from the protocol. Thank you for raising this point.

Thank you also for your comments on other criteria to consider. For the scoping review we would want to consider whether metabolic control or time of diagnosis influenced protein status, as opposed to using these as inclusion or exclusion criteria. We want to allow for consideration of all evidence of protein status in people with PKU. Both metabolic control and time of diagnosis are important factors and therefore we want to extract data on these factors to determine their impact on protein status in people with PKU. In the original manuscript, we had included ‘time of diagnosis’ in Box 1: key data to be extracted, but had not specified ‘metabolic control’. In the revised manuscript, we have updated Box 1 to include metabolic control. Furthermore, we have also amended Box 1 to include ‘genotype’ as some studies investigating protein status took into consideration participant’s genotype.

7. The authors refer “For the purpose of this review, ‘on diet’ is defined as restricting dietary protein and/or taking at least one protein substitute daily, and ‘off diet’ is defined as eating freely without restriction and not taking protein substitutes”. How do authors think they will be able to precisely identify the patients correctly, since there are frequent changes on dietary compliance in most, if not all, patient’s series? In addition, do authors think they will have access to this data?

RESPONSE: Thank you for this comment. In the ideal situation it would be useful to categorise study participants based on their dietary behaviours/patterns, but we agree that gaining access to this information could be challenging. Accordingly, we have updated the protocol to remove the specific categorisation of ‘on diet’ and ‘off diet’. Instead, we have included the following statement (Page 8, paragraph 2):

‘Although lifelong treatment is recommended in the management of PKU, some adolescents and adults may choose to have periods of time on an unrestricted diet. If available, information on participants’ history with the PKU diet will be obtained, as the impact of changes in total protein intake on and off diet on protein status warrants consideration.’

Additional comments from the authors:

- Table 1, ‘Functional’ section: This has been updated to specify other measures of functional status (i.e. isokinetic quadriceps strength using dynamometry and vertical jump performance using force platform technology)
- The search strategy has been updated to include further key terms to encompass functional measures (“VO2max” OR exp Physical Exertion/ OR "physical exertion".mp. OR exp Exercise Test/ OR "exercise test".mp.). The revised search strategy has been uploaded as supplementary material.